# Study on the Characteristics of Coherent Supersonic Jet with Superheated Steam

Xin Li [1,2], Guangsheng Wei [1,2,*], Rong Zhu [1,2,*], Bohan Tian [1,2], Ruimin Zhao [1,2] and Xinyi Lan [3]

1   The School of Metallurgical and Ecological Engineering, University of Science and Technology Beijing, Beijing 100083, China; b20200111@xs.ustb.edu.cn (X.L.); 17801054972@163.com (B.T.); m1215941648@163.com (R.Z.)
2   The Beijing Key Laboratory of Research Center of Special Melting and Preparation of High-End Metal Materials, University of Science and Technology Beijing, Beijing 100083, China
3   School of Automation and Electrical Engineering, University of Science and Technology Beijing, Beijing 100083, China; lx15932592218@126.com
*   Correspondence: wgshsteel@126.com (G.W.); zhurong12001@126.com (R.Z.)

**Abstract:** By establishing a mathematical model to simulate a mixed jet of oxygen and superheated steam from a coherent supersonic jet oxygen lance, we studied the effect of superheated steam on the fluid characteristics of the mixed jet. The model was initially verified through laboratory experiments prior to analyzing the fluid characteristics of the mixed jet in detail. These characteristics included the jet velocity, the temperature, the turbulent kinetic energy (TKE), and the mass distribution. The results showed that, at an ambient temperature of 1700 K, the jet velocity measured in the laboratory experiment was consistent with the fluid velocity obtained by numerical simulations, with an error of only 2.7%. In a high-temperature environment, the jet velocity of the mixed oxygen and superheated steam jet was increased, the TKE around the center jet was enhanced, the superheated steam exhibited an inhibitory effect on the combustion reaction of annular methane, and the potential core length of the coherent supersonic jet was reduced, which was conducive to methane combustion and delayed the reduction in the central jet velocity.

**Keywords:** coherent jet; superheated steam; numerical simulation; combustion experiment

## 1. Introduction

With the continuous efforts to enhance environmental protection and the introduction of carbon emission peak and carbon neutrality targets, short-process steelmaking has received increasing attention. However, during electric furnace smelting, large amounts of metal oxides are produced by the blowing oxygen, resulting in soot formation. This is problematic both from an environmental point of view and due to the fact that it reduces the metal yield of steelmaking. Studies have shown that the mixed injection of oxygen and carbon dioxide has an inhibitory effect on smoke and dust formation [1,2], and some domestic steel plants have begun to apply the technology of mixing oxygen and carbon dioxide in the furnace bottom or furnace wall, with obvious beneficial effects. This technique not only reduces the generation of smoke and dust but also enhances the stirring effect of the molten pool, in addition to shortening the smelting cycle. However, this technology has a number of drawbacks. Firstly, the carbon dioxide recovery technology is not yet mature, and a large number of industrial applications will inevitably be restricted due to high costs. Furthermore, the reaction of carbon dioxide with carbon is an endothermic reaction; therefore, when the carbon dioxide content is relatively high, clogging of the bottom-blowing nozzle can occur.

To shorten the smelting cycle and improve the scrap ratio and smelting efficiency, the coherent supersonic jet oxygen lance is widely used in the electric furnace smelting process. This lance is characterized by ring oxygen and ring-fired pipes around a central oxygen

pipe, thus ensuring the stability of the central jet, prolonging its potential center during the blowing process, and, thereby, achieving a good stirring effect [3].

To date, many studies have been conducted in regard to the mixed injection of coherent supersonic jet oxygen lances [4–7]. In addition, based on the vortex dissipation conceptual model, a technical jet carbon dioxide and oxygen mixed injection model was established, and the relevant characteristics of the jet were studied [2]. It was found that carbon dioxide significantly inhibited the combustion reaction of methane, and this effect was enhanced with an increase in the carbon dioxide concentration. Furthermore, as the ambient temperature increased, the potential core length of the jet also increased. The characteristics of the jet and the structure of the oxygen lance have also been examined. For example, computational fluid dynamics (CFD) software and a modified K-epsilon turbulence model were used to simulate the influence of the ambient temperature on the supersonic jet [8]. It was found that high temperatures were beneficial in terms of increasing the length of the core section of the jet. In addition, Li [9] studied the process of a nozzle jet hitting a molten pool. In this case, the jet flow volume was found to drive the movement of the slag, and the jet flow energy had a significant influence on the reaction of the molten pool. Furthermore, through a combination of CFD simulations and experiments, the characteristics of supersonic jets with and without flame coverings were studied at room temperature [10], and it was found that the potential core length of the flame-covered supersonic jet was more than three times that of a non-flame-covered supersonic jet. Moreover, Tang [11] studied the influence of the ambient temperature and the fuel composition (i.e., the blast furnace gas, natural gas, and coke oven gas) on the length of the potential core of the coherent jet. The obtained results showed that the length of the coherent jet's potential core increases with an increase in the ambient temperature, a lower fuel molecular weight, or a lower gas density, all of which increase the potential core length of the coherent jet. The influence of the traditional Laval nozzle and the curved Laval tube on the jet characteristics was also studied, and it was found that the curved Laval tube can extend the length of the core section of the jet and enhance the stirring effect of the molten pool [12]. In addition, Faheem [13] combined CFD simulations and experiments to study the characteristics of single-, double-, and triple-jet jets. The obtained data showed that the interaction between multiple jets increased the length of the core, but the height of the three jets was uneven and symmetrical until fully integrated.

Thus, we herein report the use of the mixed spraying of oxygen and water vapor to reduce the generation of smoke and dust in the furnace and to improve the production efficiency. Currently, few reports exist describing the use of steam in the steelmaking process. As an example, in the converter smelting of medium- and low-carbon steel, mixed steam injection has been found to play the role of decarburization and chromium preservation [14]. However, the mixed injection of oxygen and superheated steam in the electric furnace smelting process has yet to be reported [15]. In this study, a computational fluid dynamics model was established to simulate the flow field and characteristics of a mixed jet of oxygen and superheated steam. This model uses a modified k-$\varepsilon$ turbulence model and also considers the effects of temperature gradient and compressibility. In addition, 32-step $CH_4$-$O_2$ is used as the vortex dissipation conceptual model of the combustion reaction mechanism. This model is based on 325 basic reactions and 53 components, so it is better able to reflect a real combustion situation.

## 2. Combustion Experiments

The static pressure, total pressure, and total temperature of the mixed coherent supersonic jet were measured. Figure 1 shows the principle of the experimental measurements. The combustion furnace used in the experiment is shown in Figure 2. A water-cooled pitot tube was used to measure the static pressure and the total pressure of the jet, while a thermocouple was used to measure the total temperature. To improve the accuracy of positioning, a level gauge and a laser rangefinder were used to adjust the measurement position. The specific steps employed in this experiment were as follows: (1) The oxygen

lance was fixed to the bracket, and the corresponding pipeline was connected. (2) The pitot tube was fixed to measurement position 1, the oxygen lance switch was turned on, and the peripheral jet was ignited with a tiny flame. After the jet and combustion flame had stabilized, the measured static pressure and total pressure were considered to be effective values. (3) The oxygen lance was turned off, and the water-cooled pitot tube was replaced with a thermocouple. The total temperature of the jet was then measured at the same position. The remaining three data points were obtained by using the same procedure.

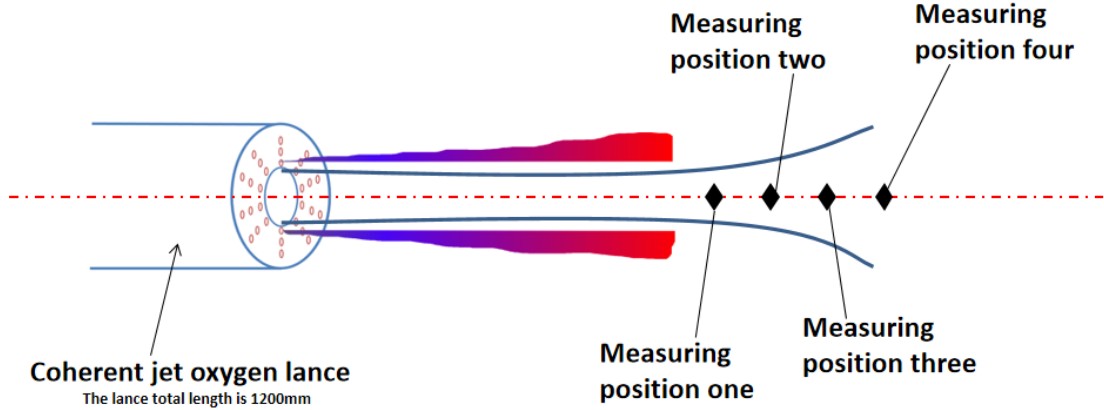

**Figure 1.** Schematic diagram of the coherent supersonic jet and the experimental measurement principle.

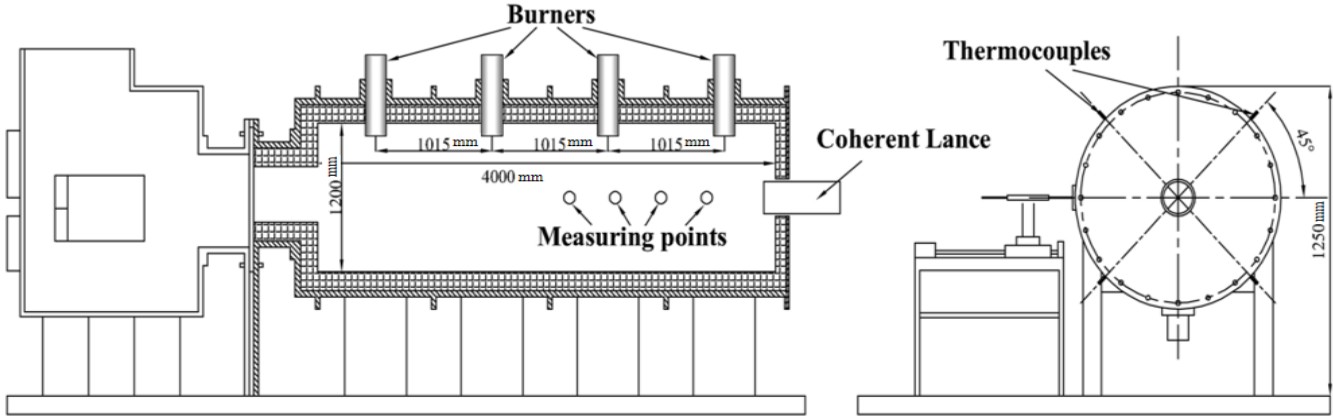

**Figure 2.** Schematic diagram of the combustion furnace.

Faheem et al. [16] studied the use of schlieren images to estimate the Mach number of a supersonic jet; however, this method cannot accurately measure the jet velocity. Therefore, the current study is based on the measured jet pressure and the fluid dynamics theory equation to allow for the calculation of the coherent supersonic jet speed as follows:

$$V^2 = \frac{2\gamma RT}{\gamma - 1}\left[\left(\frac{p_0}{p}\right)^{(\gamma-1)/\gamma} - 1\right] \tag{1}$$

where $p_0$ is the total pressure of the supersonic jet, $p$ is the static pressure of the supersonic jet, $\gamma$ is the ratio of the heat capacity, and $T$ is the static temperature of the supersonic jet. The static temperature of the jet was calculated by using the following formula:

$$T = T_0\left(\frac{1}{1 + r\frac{\gamma+1}{2}Ma^2}\right) \tag{2}$$

where $T_0$ is the total temperature of the supersonic jet, $r$ is the coefficient of the kinetic energy converted into enthalpy, and $Ma$ is the Mach number of the supersonic jet.

## 3. Numerical Simulations

The commercial software ANSYS Fluent 2019 R1 developed by ANSYS, USA was used to solve the control equations; the k-ε model was used to simulate the turbulent motion of the fluid; the species transfer model was used to simulate the transfer between the fluid and natural gas that was caused by the combustion process; and the eddy current consumption (EDC) scattered model was used to calculate the combustion reaction [17]. The one-step combustion reaction model calculates the combustion reaction between methane and oxygen, which is relatively simple and fast. However, because the reaction model only considers the final products of combustion, namely carbon dioxide and water, and does not consider intermediate products such as CO, OH, $H_2O_2$, CH, and $H_2$, the calculation results have a large error range. The combustion model used in this study was the GRI-Mech 3.0 model. This model consisted of 53 components and 325 basic reactions. It contains almost all the intermediate products of the combustion reaction between methane and oxygen and the various corresponding reactions, and so the calculation results are more aligned with the actual combustion process [18]. The DO model and the modified weighted gray gas model were used to simulate radiative heat transfer in the methane combustion process, and the control equation was solved under steady-state conditions.

### 3.1. Governing Equations

The mass conservation equation is as follows:

$$\frac{\partial \rho}{\partial t} + \nabla \cdot \left( \rho \vec{v} \right) = 0 \tag{3}$$

while the momentum conservation equation is as follows:

$$\frac{\partial}{\partial t} \left( \rho \vec{v} \right) + \nabla \cdot \left( \rho \vec{v} \vec{v} \right) = -\nabla P + \nabla \cdot \left( \overline{\overline{\tau}} \right) + \rho \vec{g} \tag{4}$$

where $P$ is the static pressure, and $\rho \vec{g}$ is the gravitational body force. In addition, $\rho$ is the fluid density, and $\vec{v}$ is the fluid velocity with x, y, and z components. The stress tensor, $\overline{\overline{\tau}}$, is related to the strain rate and is calculated by the following equation:

$$\overline{\overline{\tau}} = \mu \left[ \left( \nabla \cdot \vec{v} + \nabla \cdot \vec{v}^T \right) - \frac{2}{3} \nabla \cdot \vec{v} I \right] \tag{5}$$

where $\mu$ is the turbulent viscosity, $I$ is the unit tensor, and the second term on the right-hand side of the equation is the effect of volume dilation.

The energy equation is as follows:

$$\frac{\partial}{\partial t} (\rho E) + \nabla \cdot \left( \vec{v} (\rho E + p) \right) + \nabla \cdot \left( k_{eff} \nabla T - \sum_j h_j \vec{J}_j + \left( \overline{\overline{\tau}}_{eff} \cdot \vec{v} \right) \right) + S_h \tag{6}$$

where $\rho$ is the density, $\vec{J}_j$ is the diffusion flux of species $j$, and $S_h$ is the volumetric heat source. The first three terms on the right-hand side of the equation represent the energy transfer due to conduction, species diffusion, and viscous dissipation, respectively. $E$ is the total energy and can be calculated from Equation (7). Furthermore, $h$ is the sensible enthalpy. In this study, the gas is set as an ideal gas, and so its $h$ value is defined as in Equation (8).

$$E = h - \frac{p}{\rho} + \frac{v^2}{2} \tag{7}$$

$$h = \sum_j Y_j h_j \tag{8}$$

where $Y_j$ is the mass fraction of the species, and $h_j$ is the sensible enthalpy of species $j$, which can be calculated as follows:

$$h_j = \int_{T_{ref}}^{T} C_{p,j} dT \tag{9}$$

where $k_{eff}$ is the effective conductivity, and the definition of $k_{eff}$ is given as follows:

$$k_{eff} = k + \frac{C_p \mu_t}{Pr_t} \tag{10}$$

where $Pr_t$ is the turbulent Prandtl number, which is 0.85, the default in the ANSYS Fluent software. $C_p$ is the specific heat, and $\mu_t$ is the turbulent viscosity.

### 3.2. Standard k-ε Model

The turbulence kinetic energy transport equation is shown below [19]:

$$\frac{\partial}{\partial t}(\rho k) + \frac{\partial}{\partial X_i}(\rho k u_i) = \frac{\partial}{\partial X_j}\left[\left(\mu + \frac{\mu_t}{\sigma_k}\right)\frac{\partial k}{\partial X_j}\right] + G_k - \rho\varepsilon - Y_M + S_k \tag{11}$$

where $\rho$ is the density of the fluid, $\mu$ is the molecular viscosity, $\mu_t$ is the turbulent viscosity, $G_k$ represents the generation of turbulence kinetic energy due to the mean velocity gradients, and $\sigma_k$ is the turbulent Prandtl number for $k$ with a value of 1.0. Furthermore, $S_k$ is a user-defined source term, and $Y_M$ is modeled according to the equation proposed by Sarkar:

$$Y_M = 2\rho\varepsilon M_t^2 \tag{12}$$

where $M_t$ is the turbulent Mach number, which is defined as follows:

$$M_t = \sqrt{\frac{k}{a^2}} \tag{13}$$

where $\alpha \equiv \sqrt{\gamma R T}$ is the speed of sound.

The equation for the rate of dissipation transport is shown below:

$$\frac{\partial}{\partial t}(\rho\varepsilon) + \frac{\partial}{\partial X_i}(\rho\varepsilon u_i) = \frac{\partial}{\partial X_j}\left[\left(\mu + \frac{\mu_t}{\sigma_\varepsilon}\right)\frac{\partial\varepsilon}{\partial X_j}\right] + C_{1\varepsilon}\frac{\varepsilon}{k}G_k - C_{2\varepsilon}\rho\frac{\varepsilon^2}{k} + S_\varepsilon \tag{14}$$

where $C_{1\varepsilon}$ and $C_{2\varepsilon}$ are constants with values of 1.44 and 1.92, respectively. In addition, $\sigma_\varepsilon$ is the turbulent Prandtl number for $\varepsilon$, which has a value of 1.3. $S_\varepsilon$ is the user-defined source term, $\mu$ is the molecular viscosity, and $\mu_t$ is the turbulent viscosity, which is computed by combining $k$ and $\varepsilon$ as follows:

$$\mu_t = \rho C_\mu \frac{k^2}{\varepsilon} \tag{15}$$

where $C_\mu$ is a constant with a value of 0.09.

### 3.3. Thermal Radiation Model

Based on the calculation of the GRI-Mech 3.0 model, the methane–oxygen combustion reaction temperature was determined to be ~3000 K; therefore, radiation heat transfer must be considered. The radiation heat transfer formula is as follows [20,21]:

$$\frac{dI\left(\left(\vec{r},\vec{s}\right)\right)}{ds} + (a+\sigma_s)I\left(\left(\vec{r},\vec{s}\right)\right) = an^2\frac{\sigma T^4}{\pi} + \frac{\sigma_S}{4\pi}\int_0^{4\pi} I\left(\vec{r},\vec{s}'\right)\Phi\left(\vec{s},\vec{s}'\right)d\Omega' \tag{16}$$

where $\overrightarrow{r}$ is the position vector, $\overrightarrow{s}$ is the direction vector, $\overrightarrow{s}'$ is the scattering direction vector, $S$ is the path length, $a$ is the absorption coefficient, $n$ is the refractive index, $\sigma_s$ is the scattering coefficient, $\sigma$ is the Stefan–Boltzmann constant ($5.669 \times 10^{-8}$ W/m$^2$ K$^4$), $T$ is the local temperature, $\Phi$ is the phase function, $\Omega'$ is the solid angle, and $I$ is the radiation intensity, which depends on the position ($\overrightarrow{r}$) and direction ($\overrightarrow{s}$).

### 3.4. Calculation Area and Boundary Conditions

The computational geometry model used for this simulation is shown in Figures 3 and 4 below, including the spray gun head (Figure 3) and the combustion reaction zone (Figure 4). It is worth noting that, in the simulation, the central section of the nozzle and the computational domain is selected as the observation surface. The following discussion is also based on the observation plane. The spray gun head of the coherent supersonic jet was composed of a central Laval tube and three layers of annular seams. The three layers of annular seams are composed of 10 small holes corresponding to one another, and the central Laval tube sprays oxygen and water vapor with different mixing ratios. Two different types of gases were injected into the ring seams. Among them, Rings 1 and 3 were injected with oxygen, and Ring 2 was injected with fuel gas, namely CH$_4$, which was enhanced by the low-density space generated by the combustion reaction of oxygen and methane. The length of the potential core of the central jet was measured. The combustion reaction zone is a cylindrical area with a diameter of 700 mm and length of 2450 mm (see Figure 4). Moreover, the mesh with 201,484 cells was used in the simulation.

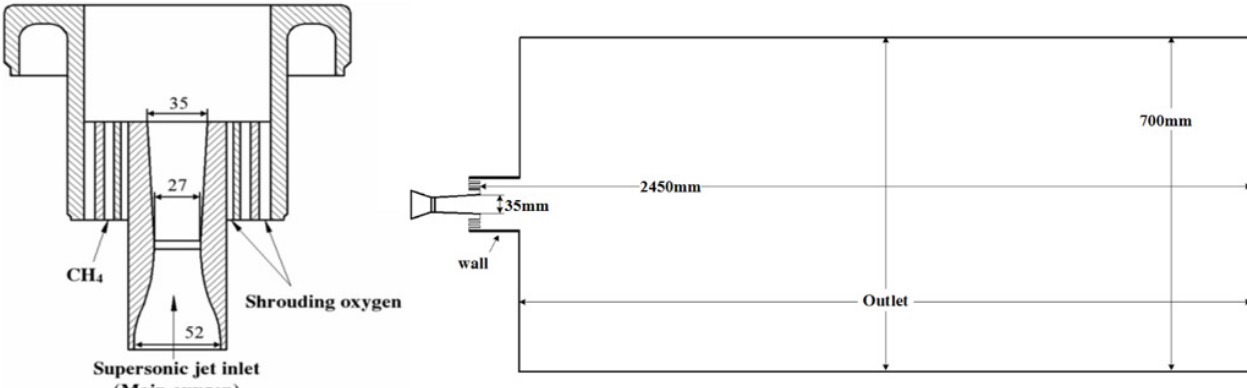

**Figure 3.** Cross-sectional view of the coherent jet nozzle and the jet-flow domain used for modeling.

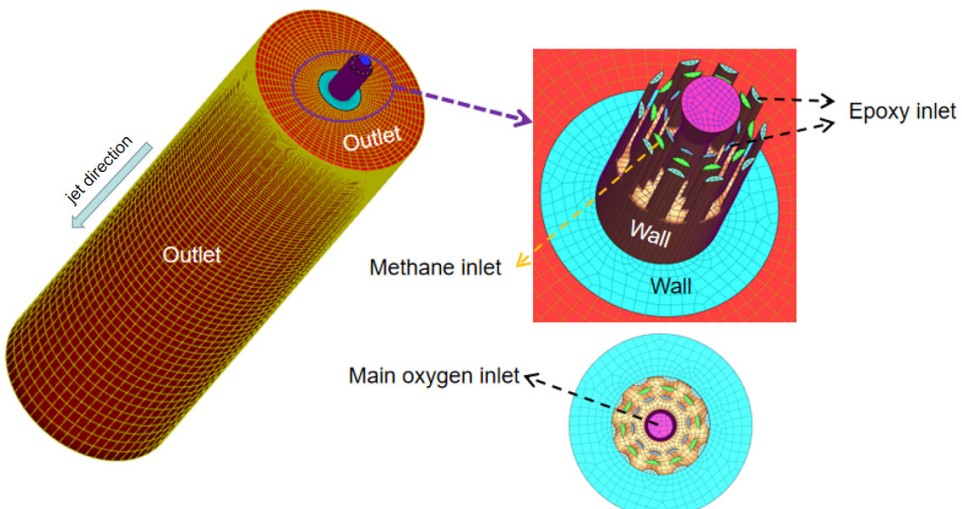

**Figure 4.** Schematic diagram of the coherent supersonic jet nozzle.

The mass flow inlet boundary condition was set at the entrance of the Laval nozzle, wherein oxygen and fuel gas cover the nozzle. The pressure boundary condition at the exit position of the combustion zone was used, as was the SIMPLE algorithm for the pressure-velocity coupling scheme. The second-order upwind formula was used to solve the momentum equation and the mass equation, wherein the convergence standards for energy, continuity, and the other dependent variables were set to $10^{-6}$, $10^{-7}$, and $10^{-5}$, respectively. The wall adopts a non-slip state and is equipped with standard wall functions. Detailed information regarding the boundary conditions and the central gas mixture is presented in Tables 1 and 2.

**Table 1.** Boundary conditions.

| Parameters | Values |
|:---:|:---:|
| Laval nozzle | |
| Nozzle Convergent Length (mm) | 32 |
| Nozzle Divergent Length (mm) | 46 |
| Throat Diameter ($d_t$) (mm) | 27 |
| Exit Diameter ($d_e$) (mm) | 35 |
| Inlet Diameter (di) (mm) | 52 |
| Designed Mach Number at Nozzle Exit | 2.0 |
| Main Oxygen Inlet | |
| Mass Flow Rate | 1.0714 kg/s |
| Total Temperature | 600 K |
| Volume Fractions | Shown in Table 2 |
| Shrouding Oxygen Inlet | |
| Mass Flow Rate | 0.079365 kg/s |
| Total Temperature | 298 K |
| Volume Fractions | $O_2$: 100 pct |
| Fuel Gas Inlet | |
| Mass Flow Rate | 0.079365 kg/s |
| Total Temperature | 298 K |
| Volume Fractions | $CH_4$: 100 pct |
| Outlet | |
| Static Pressure | 101,325 Pa |
| Wall | |
| No-Slip, Fixed Temperature | 298 K |

**Table 2.** Experimental scheme of main gas composition.

| Label | Volume Flow Rate (Nm$^3$/h) | Mass Flow Rate (kg/s) | Volume Fraction (Pct) | |
|:---:|:---:|:---:|:---:|:---:|
| 0 pct $H_2O$ | 2700 | 1.0714 | $O_2$ | 100 |
| | | | $H_2O$ | 0 |
| 25 pct $H_2O$ | 2700 | 0.9542 | $O_2$ | 75 |
| | | | $H_2O$ | 25 |
| 50 pct $H_2O$ | 2700 | 0.8371 | $O_2$ | 50 |
| | | | $H_2O$ | 50 |
| 75 pct $H_2O$ | 2700 | 0.7199 | $O_2$ | 25 |
| | | | $H_2O$ | 75 |

## 4. Results and Discussion

### 4.1. Velocity Distribution

Figure 5 shows the axial velocity distribution on the centerline of coherent supersonic jet at different temperatures, and x = 0 is the nozzle exit plane. As shown in the figure, as the $H_2O$ content increased, the speed of the coherent supersonic jet gradually increased. It should be noted here that, at 373.15 K, the density of saturated water vapor at standard atmospheric pressure is 0.6 g/NL, and its relative molecular mass is 18; the density and relative molecular mass of $O_2$ are 1.429 g/NL and 32, respectively. When the total gas

volume and stagnant pressure remained unchanged, the $H_2O$ content increased. Furthermore, the lower the mass flow of the gas, the higher the relative jet velocity, and when the velocity reaches its maximum value, it begins to drop and fluctuate. The Laval spear head was designed under normal temperature and pressure conditions, in addition to full oxygen conditions. Therefore, when mixed with $H_2O$, which has a relatively low density, it reaches the stagnation required for the same velocity. The pressure was relatively small, and the outlet pressure of the lance was greater than the ambient pressure, thereby resulting in the incorrect expansion of the jet gas, which was accompanied by the corresponding fluctuations. The distance required for the jet exit velocity to decay to 95% of its original value corresponds to the core length of the jet potential.

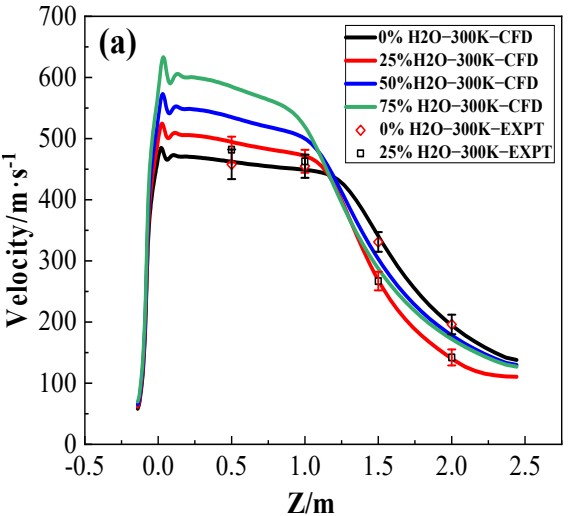 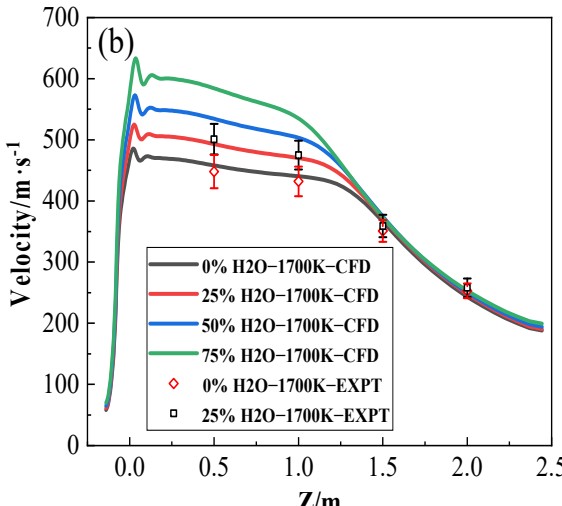

**Figure 5.** The axial velocity distribution on the centerline of coherent supersonic jet. (**a**) The velocity measured by numerical simulations and experimentally at 300 K. (**b**) The velocity value measured by numerical simulations and experimentally at 1700 K.

As shown in Figure 5a, as the $H_2O$ content was increased from 0 to 75%, the potential core length decreased from 0.911 0.481 m. In addition, as shown in Figure 5b, at an ambient temperature of 1700 K, the core length of the central jet also decreases with an increase in the $H_2O$ content. However, this decrease is obviously weakened at a relatively low ambient temperature. For the jet core with a high ambient temperature, the length is greater than that of the jet potential core at a low ambient temperature. This indicates that a high-temperature environment is beneficial to extending the length of the jet core, the density of the gas in the free space in a high temperature environment is low, and the kinetic energy loss during the jet operation is small. As a result, the jet can maintain high-speed movement over a long distance.

By combining the results presented in Figure 5 and Table 3, it can be seen that the jet velocity obtained by the CFD simulation is comparable to the jet velocity measured under experimental conditions. At an ambient temperature of 300 K, the error between the two was 1.6%, and when the ambient temperature was 1700 K, the error value increased to 2.7%, which was mainly attributed to the high-temperature influence of the measuring instrument after maintaining the combustion furnace at a high temperature for a long period of time; this led to an increase in the experimental measurement error.

In order to determine the influence of the surrounding combustion on the jet, the flow field calculation without combustion was carried out and compared with the jet with combustion. The results are shown in Figure 6. It can be seen from the figure that, with the addition of 75% steam, the jet velocity increases, and the jet is extended, indicating that adding water vapor is beneficial to the jet; the combustion of the surrounding oxygen is also beneficial to the enhancement of the jet. At the same time, due to the low-density area generated by the combustion, the central jet is diffused outward, thereby increasing the

overall jet coverage, which will be more conducive to reducing the smoke and dust in the smelting process.

**Table 3.** Numerical simulation and experimental measurement of jet velocity values at ambient temperatures of 300 and 1700 K.

| | Ambient Temperature 300 K | | | Ambient Temperature 1700 K | | |
|---|---|---|---|---|---|---|
| Z/mm | CFD m/s | EXPT m/s | Deviation/% | CFD m/s | EXPT m/s | Deviation/% |
| 500 | 462 | 458 | 0.8 | 458 | 448 | 2.2 |
| | 494 | 482 | 2.4 | 493 | 501 | 1.6 |
| 1000 | 449 | 455 | 1.3 | 441 | 432 | 2.1 |
| | 471 | 463 | 1.7 | 470 | 475 | 1.0 |
| 1500 | 335 | 331 | 1.2 | 363 | 351 | 3.4 |
| | 260 | 267 | 2.6 | 371 | 359 | 3.3 |
| 2000 | 193 | 196 | 1.5 | 244 | 253 | 4.3 |
| | 140 | 142 | 1.4 | 248 | 258 | 3.9 |
| Average | | | 1.6 | | | 2.7 |

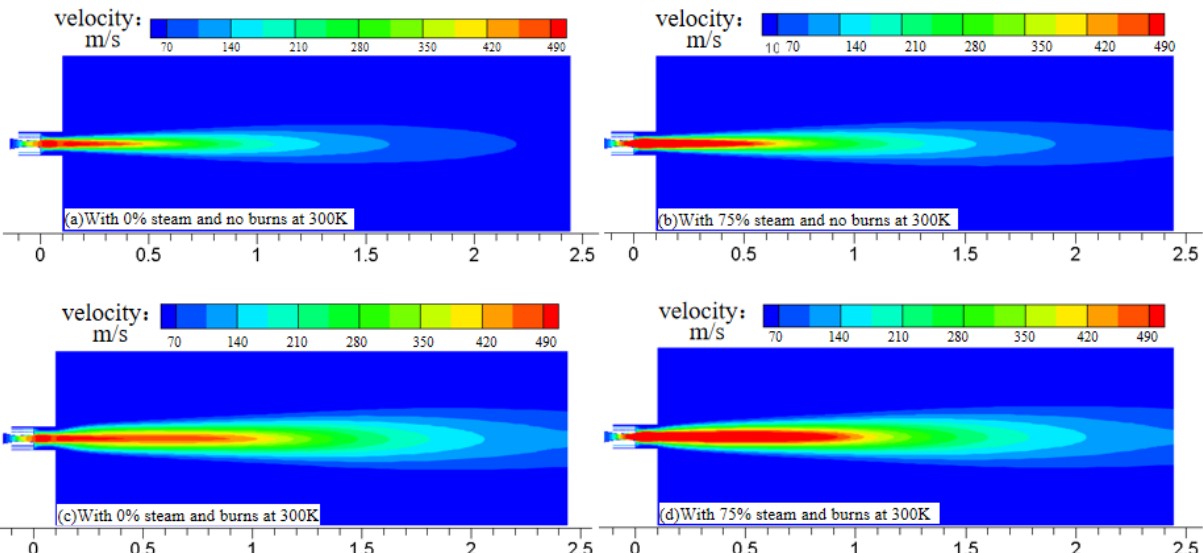

**Figure 6.** Axial velocity nephogram on the centerline of a coherent supersonic jet.

The nozzle in Figure 6 is further enlarged, and the result is shown in Figure 7. It can be seen from Figure 7 that the jet velocity at the nozzle has obvious fluctuations, which are shown in the fluctuation section of the velocity in Figure 5. The reason is explained in the above analysis. In order to better compare the velocity of the jet in different situations, the scales in the figure are unified, so the fluctuation phenomenon is masked in Figure 7b,d.

Figure 8 shows the density distribution of the coherent supersonic jet in free space when the ambient temperature was 300 and 1700 K and the oxygen content was 100%. It can be seen that, in a high-temperature environment, the gas density in the free space is significantly lower than that in the normal temperature environment, and the center jet appears on both sides; in addition to the low-density space produced by the combustion of ring-slit methane, the existence of a low-density space reduces the growth rate of the turbulent mixing zone. If the growth rate of the turbulent mixing zone decreases, the velocity attenuation of the central jet decreases. Based on the above analysis, it can therefore be concluded that the addition of $H_2O$ has an inhibitory effect on the combustion of methane in the annular joint. However, under a high ambient temperature, it is conducive to the combustion of methane, and so the high-temperature environment is beneficial to

the extension of the jet potential core; this can also explain the abovementioned jet axial velocity distribution.

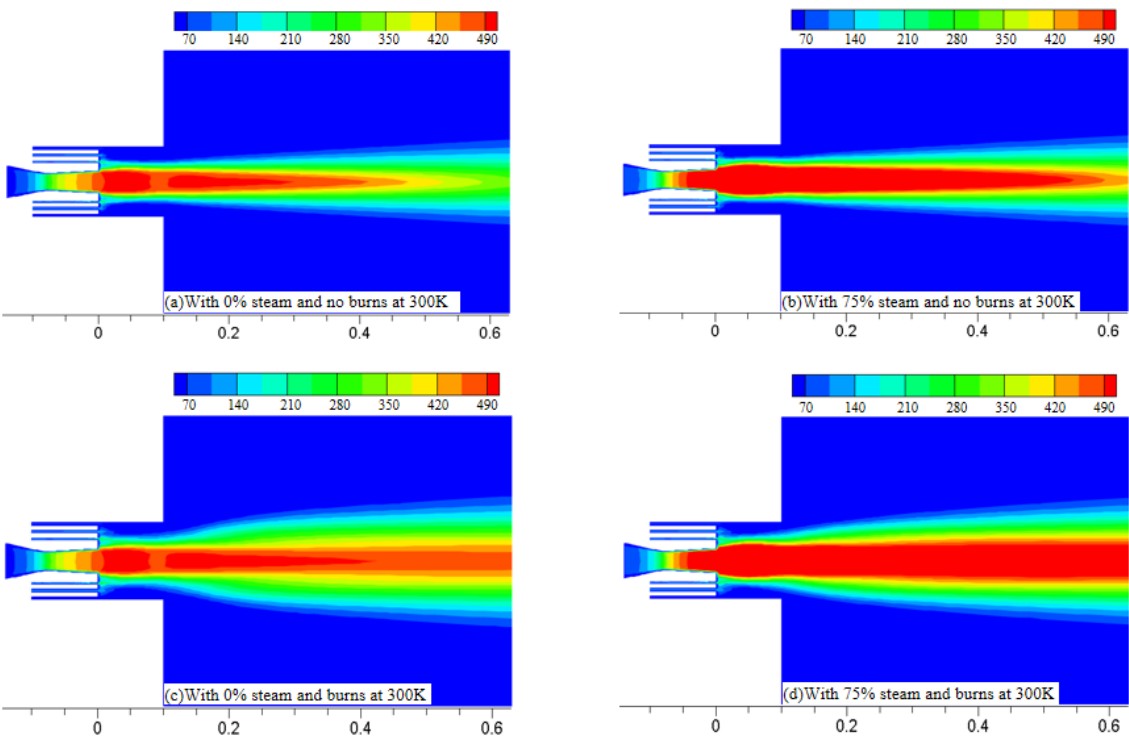

**Figure 7.** Enlarged view of the axial velocity nephogram on the centerline of the coherent supersonic jet.

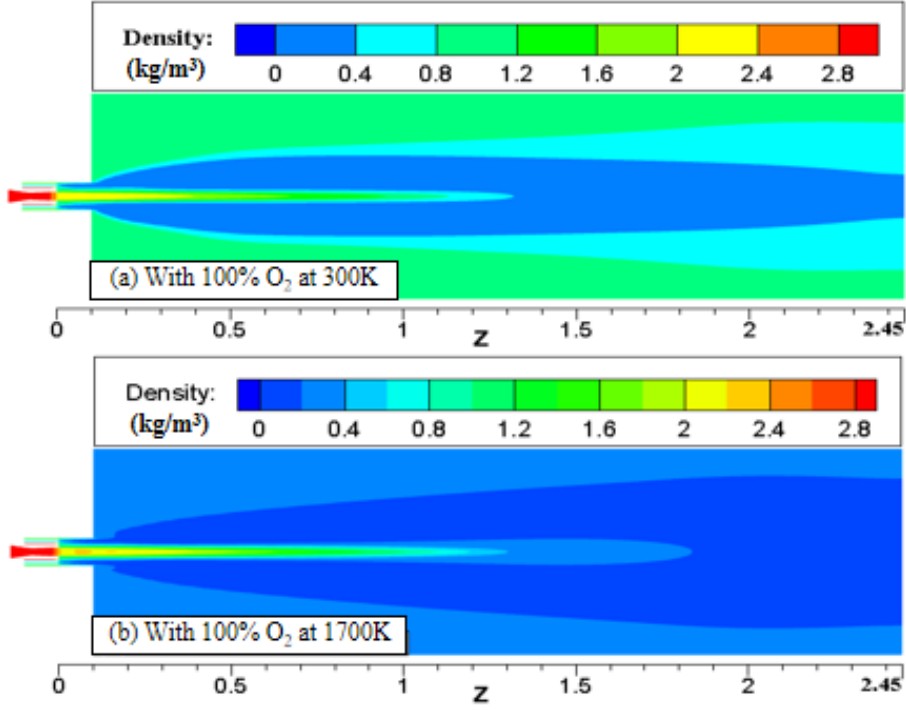

**Figure 8.** Density distribution of the full oxygen coherent supersonic jet at 300 and 1700 K. (**a**) The density distribution of the mixed injection jet at an ambient temperature of 300 K. (**b**) The density distribution of the mixed injection jet at an ambient temperature of 1700 K.

### 4.2. Temperature Distribution

Figure 9 shows the total temperature distribution in the axial direction under different water vapor contents at ambient temperatures of 300 and 1700 K. As shown in Figure 9a, after exiting the spray gun, the total temperature of the jet remained constant within a certain distance and then rapidly increased, prior to slowly dropping to room temperature after reaching a peak. With an increase in the water vapor content, the maximum temperature gradually decreased, and the temperature began to increase at a gradually lower rate; the jet temperature measured in the laboratory was compared to that obtained by the CFD simulation. In addition, Figure 9b shows that, after the jet exits from the spray gun, its total temperature is initially maintained over a certain distance, and then rises rapidly prior to gradually stabilizing close to the ambient temperature. As the water vapor mixing ratio increases, the total rate of the jet temperature increase gradually slows down, the water vapor has an inhibitory effect on the combustion of methane, and the heat of combustion decreases, ultimately leading to the jet heating rate slowing down and the maximum temperature dropping. The curve of the initial temperature rise shown in Figure 9b was enlarged and is shown in the neutron diagram in Figure 9b. From this inset, it can be seen that the increase in the mixing amount of water vapor causes the total temperature of the jet to rise earlier, which is mainly due to the fact that the specific heat capacity of high-pressure water vapor is lower than the specific heat capacity of oxygen. Therefore, the higher the proportion of water vapor, the slower the rate of temperature rise at the same temperature.

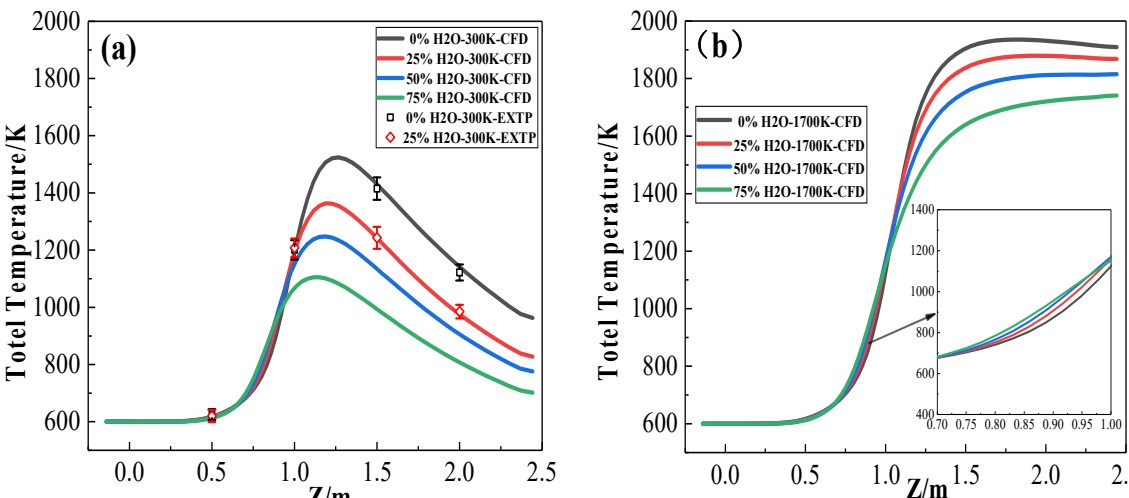

**Figure 9.** The total temperature distribution of the $H_2O$ and $O_2$ mixed injection jet along the axis. (**a**) The temperature distribution of the mixed injection jet with different components at an ambient temperature of 300 K. (**b**) The temperature distribution of the mixed injection jet with different components at an ambient temperature of 1700 K.

Figures 10 and 11 show the total temperature distribution on the longitudinal interface of the jet with different water vapor contents when the ambient temperature was 300 and 1700 K. The red regions in these figures represent the central high-temperature zone produced by the combustion of annular gas and oxygen. Upon increasing the water vapor content, the observed shrinkage of the central high-temperature zone indicates that the reaction between methane and oxygen is inhibited by water vapor; this is consistent with the results presented in Figure 9. By comparing the central high-temperature zones at 300 and 1700 K, it is apparent that higher ambient temperatures result in larger central high-temperature zones, as generated by the combustion reaction. This is mainly due to the relatively low gas density in the free space at a high ambient temperature, in addition to the annular gas flow and the central gas flow, which easily diffuse outward. The low environmental density at high temperatures also prolongs the turbulent mixing time of the

jet flow and the external gas. Therefore, the high-temperature zone of the jet center in a high-temperature environment is larger and gradually deviates from the axis as the axial distance increases.

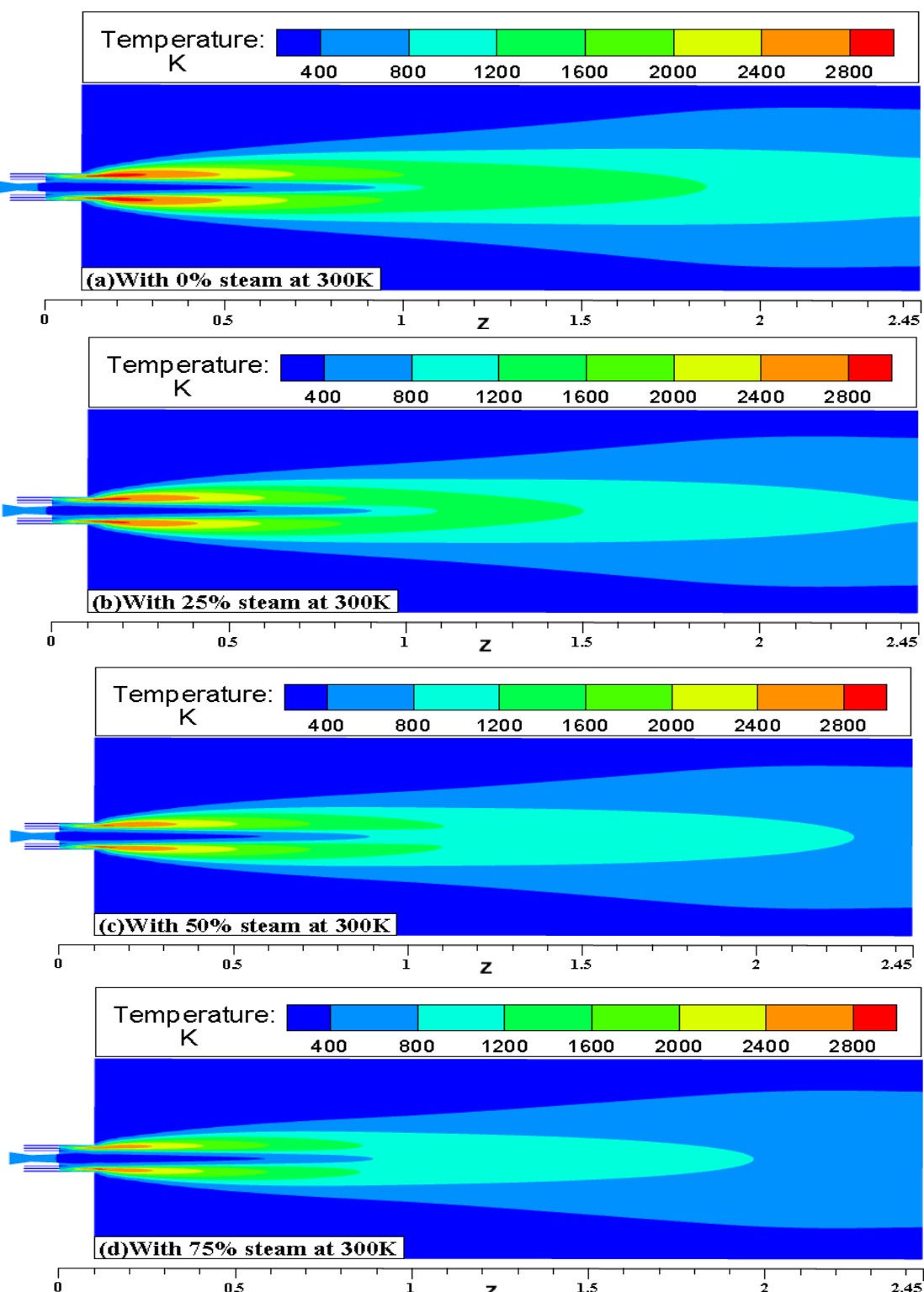

**Figure 10.** Total temperature distributions of the $H_2O$ and $O_2$ mixed jets in the longitudinal section at an ambient temperature of 300 K under various conditions: (**a**) 0% steam and 100% oxygen; (**b**) 25% steam and 75% oxygen; (**c**) 50% steam and 50% oxygen; and (**d**) 75% steam and 25% oxygen.

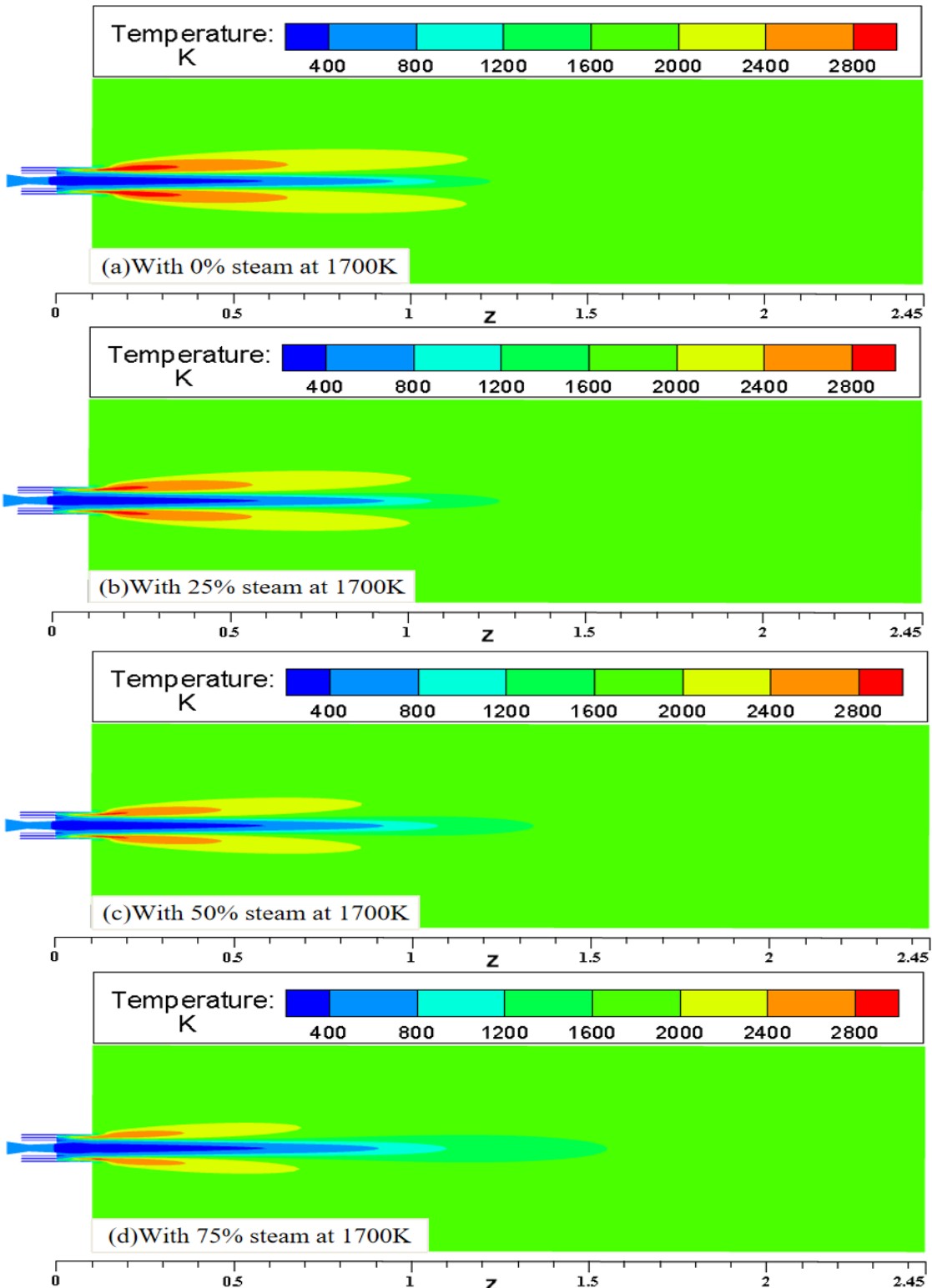

**Figure 11.** Total temperature distributions of the H$_2$O and O$_2$ mixed jets in the longitudinal section at an ambient temperature of 1700 K under various conditions: (**a**) 0% steam and 100% oxygen; (**b**) 25% steam and 75% oxygen; (**c**) 50% steam and 50% oxygen; and (**d**) 75% steam and 25% oxygen.

*4.3. Vorticity and the Turbulent Kinetic Energy Distribution*

Vorticity is known to reflect the degree of fluid mixing, wherein a greater vorticity reflects a higher degree of fluid mixing [22]. Thus, Figure 12 shows the radial vorticity

distributions of the $O_2$ and $H_2O$ mixed jets under different conditions when Z = 0.15, 0.3, 0.45, and 0.6 m. It can be seen from the figure that each vorticity curve has two peaks, wherein the peak near the centerline of the jet is caused by the interaction between the peripheral oxygen and the oxygen in the center jet. As the distance to the jet exit increases, the jet speed decreases, and so the vorticity also decreases. As a result, peaks are formed at a greater distance from the centerline of the jet through the combustion reaction between the annular gap $CH_4$ and the peripheral oxygen, ultimately leading to the peaks at a greater distance from the centerline becoming wider as the distance from the lance exit increases.

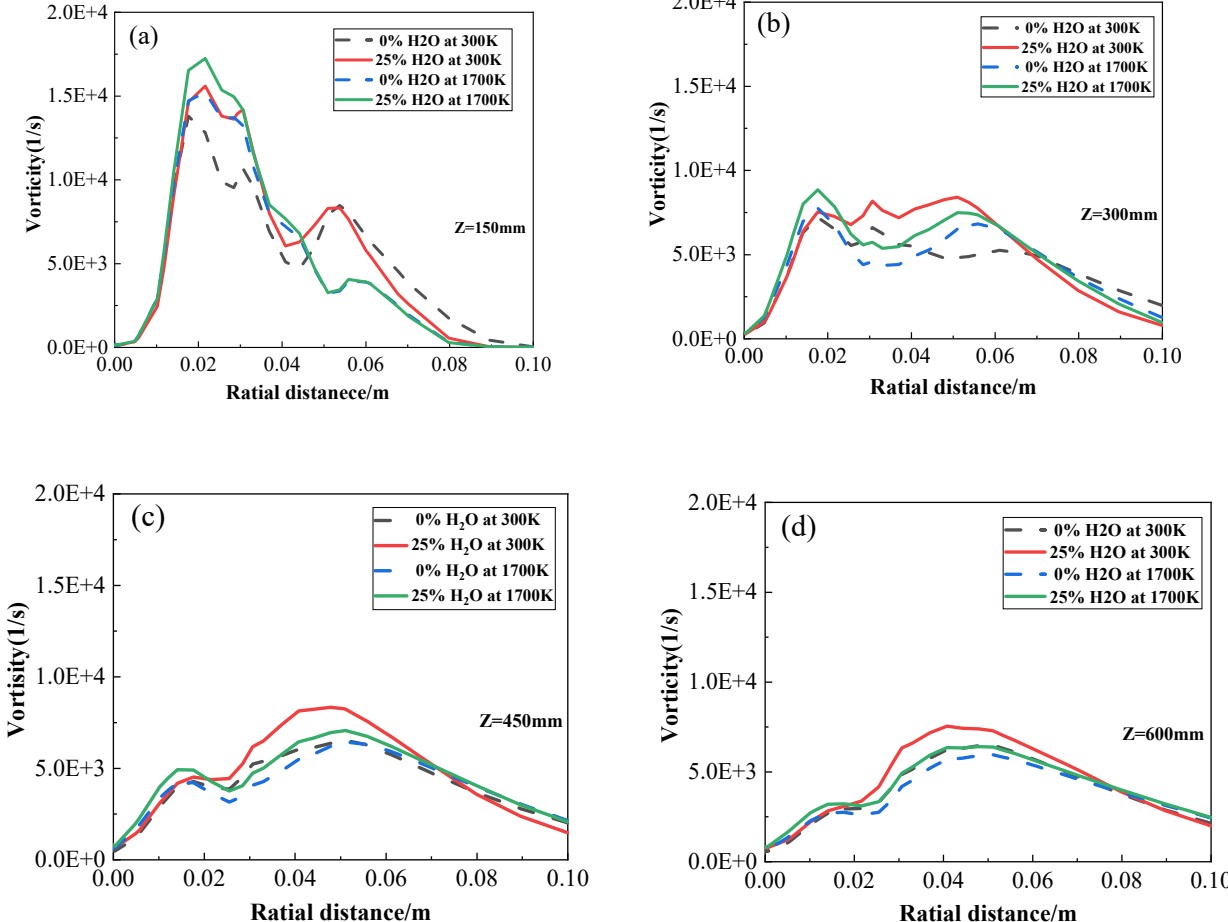

**Figure 12.** Radial distribution of the jet vorticity at different positions from the jet outlet under ambient temperatures of 300 and 1700 K: (**a**) Z = 150 mm, (**b**) Z = 300 mm, (**c**) Z = 150 mm, and (**d**) Z = 600 mm.

Figure 12b shows that the rising rate of vorticity in the high-temperature environment is higher for the first peak, indicating that the jet has a longer potential core in a high-temperature environment; because the high-temperature environment can decrease the growth rate and, conversely, the shear layer in the combustion zone, represented by the second vorticity slope peak, both high- and low-temperature environments result in the peak corresponding to the 25% water vapor condition being higher than that of the total oxygen condition. Moreover, the gradient of this peak is closer to the central axis. Under these conditions, the velocity gradient becomes larger, thereby giving a large vorticity, and so it becomes clear that the addition of water vapor inhibits the combustion reaction of the annular gas. This result has been analyzed previously and has led to advances in the mixing of the jet and the ambient gas. After the addition of $H_2O$, the second peak of the vorticity curve becomes closer to the central axis.

As reported previously, the turbulent kinetic energy (TKE) can reflect the mixing degree of the fluid to a certain extent, and the obtained TKE cloud chart can more clearly indicate the mixing degree of the jet and the surrounding gas. Thus, Figure 13 shows the TKE distribution on the longitudinal section of the coherent supersonic jet with different $H_2O$ contents at different ambient temperatures. It can be seen that, at an ambient temperature of 300 K and an $H_2O$ content of 25%, the length of the low TKE region corresponding to the central jet was the shortest, the high TKE region was the longest, and the highest TKE value was obtained (i.e., the yellow and red areas shown in Figure 13b). These results indicate that the $H_2O$ present in the center jet has an inhibitory effect on combustion, and the addition of water vapor can increase the speed of the center jet. This is consistent with the results of the above analysis. Moreover, by comparing the TKE of 300 K and 1700 K with 0% water vapor, it can be found that the turbulent mixing layer, at a high temperature, merges with the jet centerline more closely, and this is mainly due to the relatively low density of ambient gas in the high-temperature environment and the combustion of gas around the central jet. The latter is more inclined to diffuse into the surrounding environment, which delays the kinetic energy transfer between the central jet and the surrounding combustion gases, thereby lengthening the potential core.

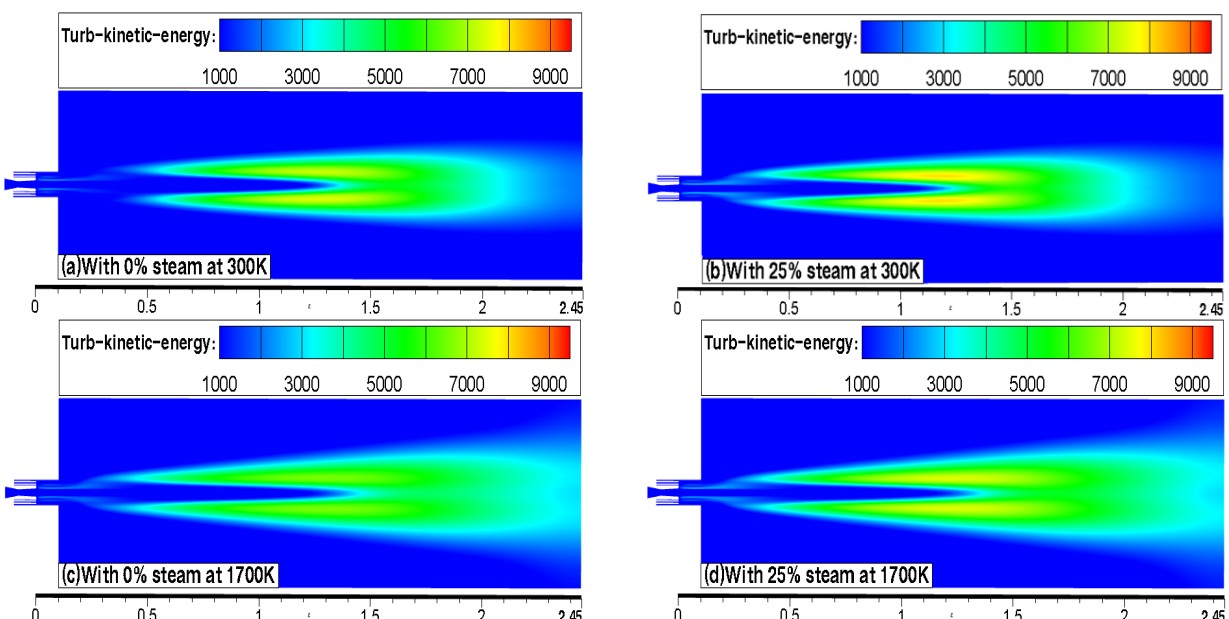

**Figure 13.** Axial cross-sectional distributions of the turbulent flow energy of the coherent supersonic jet at different temperatures and different water contents: (**a**) 0% steam and 100% oxygen at 300 K; (**b**) 25% steam and 75% oxygen at 300 K; (**c**) 0% steam and 100% oxygen at 1700 K; and (**d**) 25% steam and 75% oxygen at 1700 K.

Furthermore, Figure 14 shows the turbulent viscosity distribution of the coherent supersonic jet. In turbulent motion, the vortex drives the gas molecules to move irregularly, resulting in a strong momentum transfer rate between the gases and a change in the apparent viscosity of the fluid; this is the turbulent viscosity, which can also reflect the degree of gas mixing. As shown in Figure 14, the turbulent viscosity of the jet at a high ambient temperature is significantly lower than that at a normal temperature. Similarly, due to the relatively high density of the surrounding gas in the low-temperature environment, when the high-speed airflow in the center meets the surrounding gas, more kinetic energy will be transferred out, resulting in a faster growth rate of the shear layer. Therefore, in the low-temperature environment, the turbulent viscosity is relatively high.

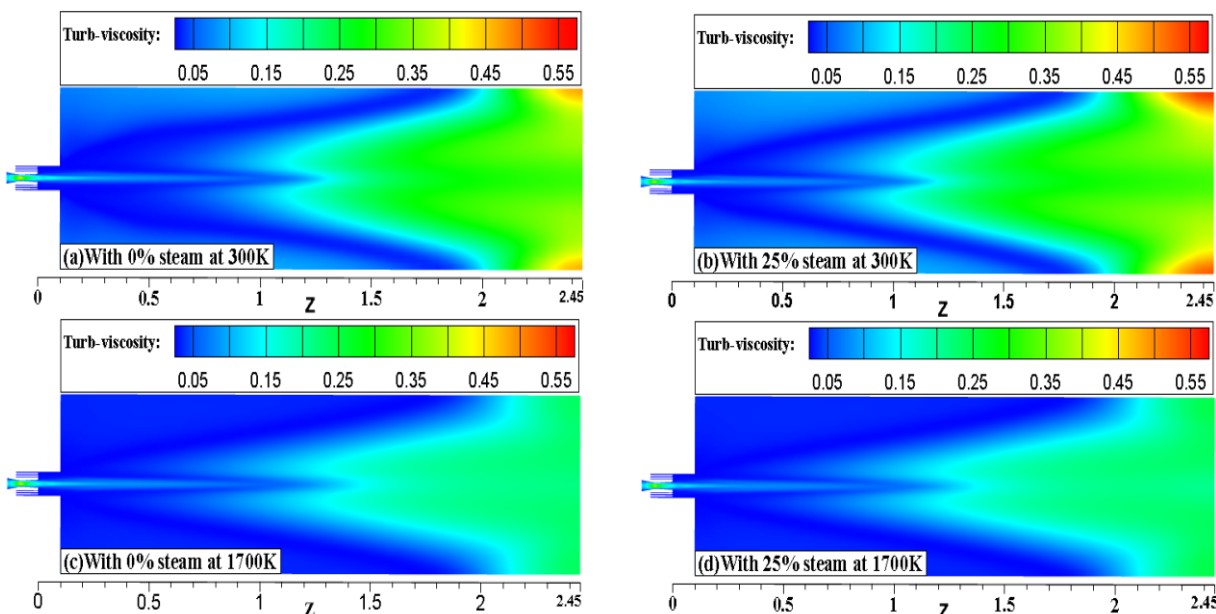

**Figure 14.** Axial cross-sectional distribution of the turbulent viscosity of the coherent supersonic jet at different temperatures and different water contents: (**a**) 0% steam and 100% oxygen at 300 K; (**b**) 25% steam and 75% oxygen at 300 K; (**c**) 0% steam and 100% oxygen at 1700 K; and (**d**) 25% steam and 75% oxygen at 1700 K.

*4.4. Material Mass Distribution*

Figure 15 shows the radial distribution of $H_2O$ in the coherent supersonic jet at Z = 150, 300, 450, and 600 mm, wherein, for each curve, a crest is also observed that corresponds to the formation of $H_2O$ and $CO_2$ via combustion. Therefore, the position of the wave crest represents the methane combustion area. The differences between the various curves can be related to the addition of different quantities of water vapor compared to the case of full oxygen. In addition, in all cases, a clear trough is observed close to the *y*-axis. More specifically, the central jet containing 25% $H_2O$ maintained a high value at a distance close to the axis. As the distance from the axis increased, the amount of diffused $H_2O$ decreased gradually until reaching the annular seam oxygen and $CH_4$ combustion zone edge, which accounts for trough formation. When the gas in the coherent supersonic jet was composed of oxygen alone, the peak value at an ambient temperature of 1700 K was lower than that at 300 K, as shown in Figure 15b–d; this was mainly attributed to the air present in the reaction environment at high temperatures. More specifically, the density of this air is relatively low, and so the outermost annular gap oxygen diffuses to a greater extent, resulting in reduced combustion at the outer layer, and ultimately leading to lower levels of generated $CO_2$ and a reduced peak maximum. Upon mixing the coherent supersonic jet with 25% $H_2O$, the peak observed at an ambient temperature of 1700 K was lower than that at 300 K, although it gradually reached higher values than the peak at 300 K as the distance from the axis increased. This is consistent with the outward diffusion of the central $H_2O$.

In addition, it can be seen from Figure 15c that, at 300 K, the peak of the total oxygen curve for the central jet is higher than that of the 25% $H_2O$ curve. At this point, the reaction of oxygen and methane plays a major role, and the presence of water vapor has a detrimental effect on the peripheral combustion; at 1700 K, the peak value of the total oxygen curve of the central jet is lower than that obtained in the presence of 25% $H_2O$, due to the role of diffusion. Although the water vapor present in the center inhibits the combustion of methane, the peak value obtained following the addition of 25% $H_2O$ remained higher than that corresponding to the pure oxygen conditions.

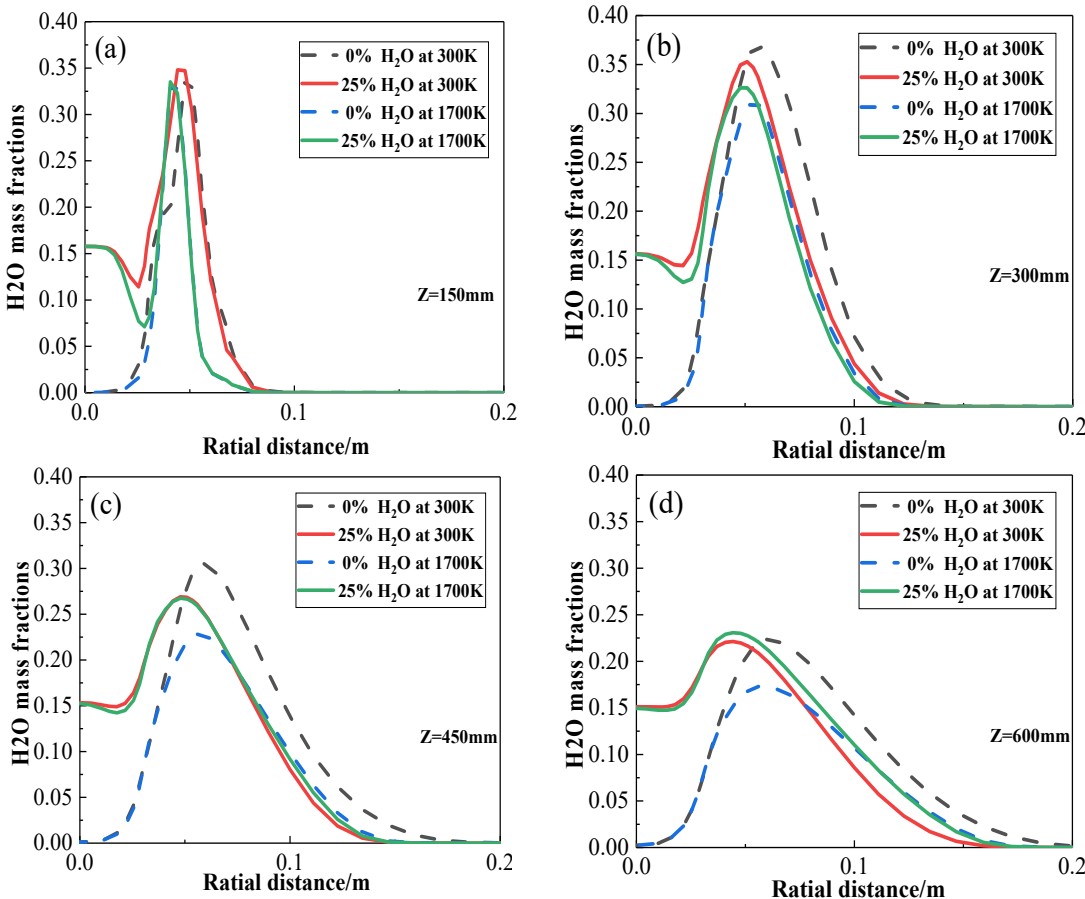

**Figure 15.** Radial distributions of $H_2O$ at different positions from the spray gun outlet at 300 and 1700 K: (**a**) Z = 150 mm, (**b**) Z = 300 mm, (**c**) Z = 150 mm, and (**d**) Z = 600 mm.

The red areas shown in Figure 14 correspond to the core area of the combustion reaction, where the $H_2O$ mass concentration is the highest. Upon comparison between Figure 16a,b, it can be seen that the combustion reaction of methane is weakened after the addition of 25% $H_2O$ to the feed; this result is consistent with the results presented in Figure 13. In addition, comparing Figure 16a,c, it is apparent that the total length of the jet corresponding to the mass distribution of $H_2O$ is significantly longer at 1700 K, thus indicating that a high ambient temperature is beneficial to the combustion reaction.

Figure 17 shows the radial distribution of $CO_2$ in the coherent supersonic jet when Z = 150, 300, 450, and 600 mm, where it can be seen that the central water vapor exhibits an inhibitory effect on the methane combustion reaction in the annular joint. More specifically, the peak values of the curves for a full oxygen flow (Figure 17d) are higher than those obtained for the 25% $H_2O$ curves. This can be attributed to the inhibitory effect of $H_2O$ on the combustion of $CH_4$, which results in a reduction in the amount of $CO_2$ generated. At 300 K, it was observed that the addition of 25% $H_2O$ to the coherent supersonic jet resulted in a reduction in the corresponding peak value compared to that of the pure oxygen peak, and a similar result was observed at 1700 K. This was attributed to the fact that, at 300 K, the jet diffuses outward to a lesser extent. In addition, since the gas density in the medium is relatively high at this temperature, the amount of $CO_2$ generated by the combustion process is higher. Following the addition of $H_2O$, the combustion reaction was suppressed, and so the decrease was more obvious. In contrast, at 1700 K, the density of the ambient gas is reduced, and the jet diffuses outward. Although a high temperature is beneficial for the combustion reaction, the amount of oxygen is insufficient to achieve the full combustion of $CH_4$. Following the addition of $H_2O$, this effect is less pronounced, and so the observed decrease in $CO_2$ production is relatively low.

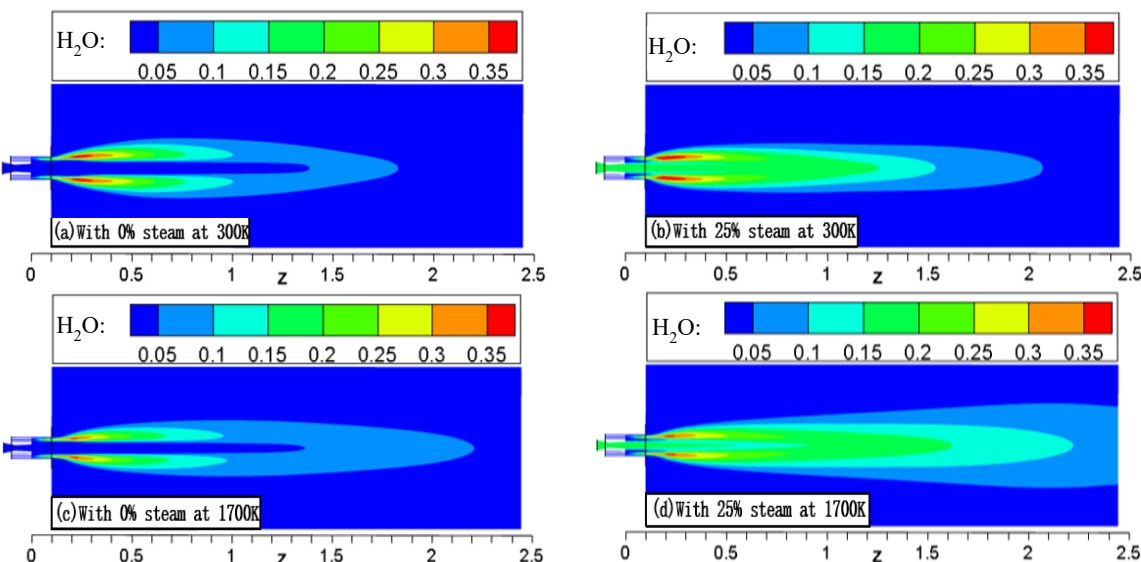

**Figure 16.** Longitudinal cross-sectional distributions of $H_2O$ in the coherent supersonic jets under different conditions: (**a**) 0% steam and 100% oxygen at 300 K; (**b**) 25% steam and 75% oxygen at 300 K; (**c**) 0% steam and 100% oxygen at 1700 K; and (**d**) 25% steam and 75% oxygen at 1700 K.

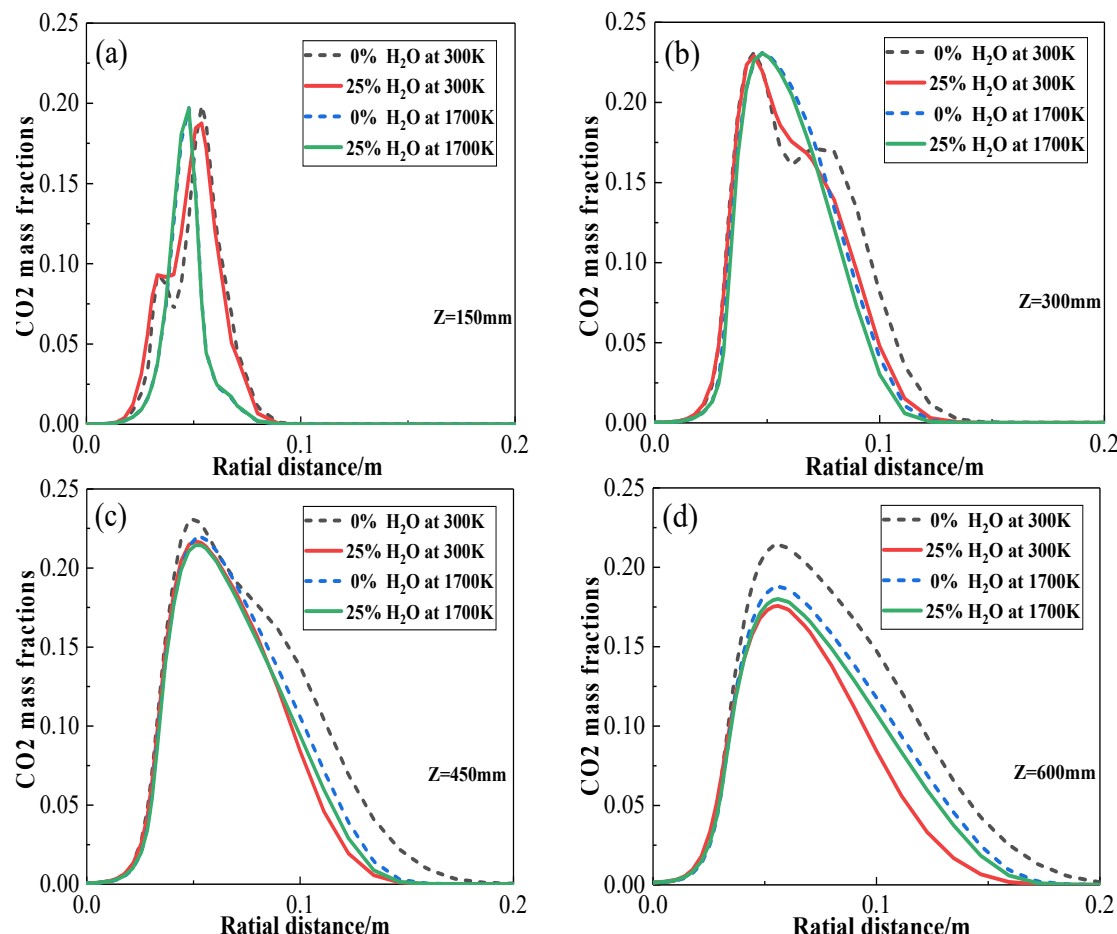

**Figure 17.** Radial distribution of the $CO_2$ mass fraction at different positions from the spray gun outlet at 300 and 1700 K: (**a**) Z = 150 mm, (**b**) Z = 300 mm, (**c**) Z = 150 mm, and (**d**) Z = 600 mm.

Finally, Figure 18 shows the longitudinal cross-sectional distribution of the $CO_2$ mass in the jet at 300 and 1700 K. It can be seen from this figure that the yellow area representing a higher $CO_2$ content in the jet is the largest under 300 K full oxygen conditions, and this yellow area clearly decreases in size following the addition of $H_2O$ at 300 K, but not at 1700 K; this is consistent with the above analysis.

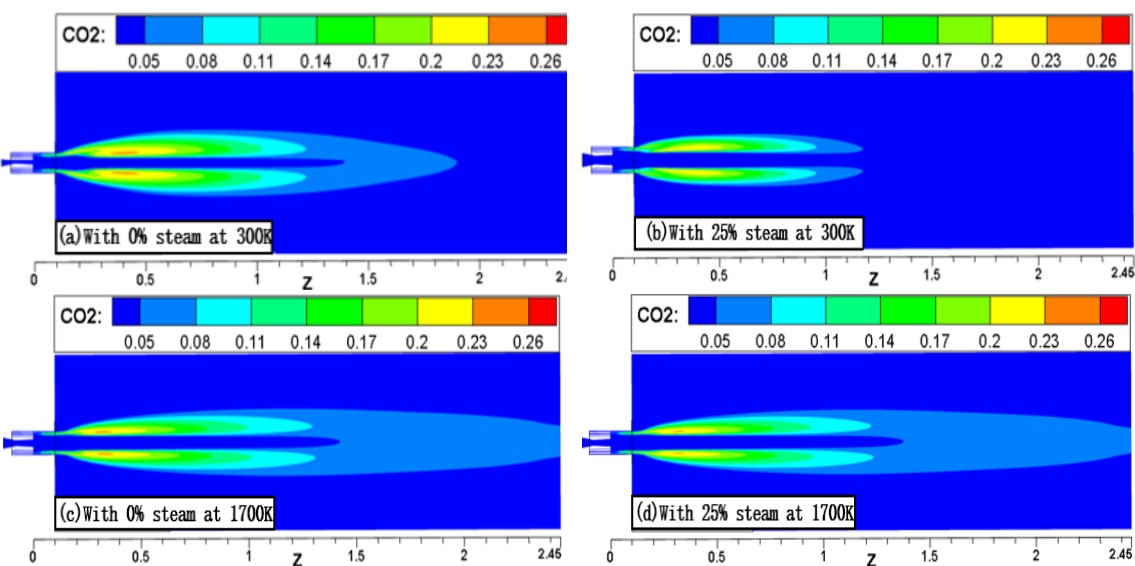

**Figure 18.** Longitudinal cross-sectional distributions of $CO_2$ in the coherent supersonic jets under different conditions: (**a**) 0% steam and 100% oxygen at 300 K; (**b**) 25% steam and 75% oxygen at 300 K; (**c**) 0% steam and 100% oxygen at 1700 K; and (**d**) 25% steam and 75% oxygen at 1700 K.

## 5. Conclusions

A mathematical model for the mixed injection of $O_2$ and superheated steam with a coherent supersonic jet oxygen lance was established, and the relevant characteristics of the superheated steam jet were analyzed. It was found that the relative mass flow of the jet was reduced upon the addition of superheated steam, and the stagnation pressure remained unchanged. As a result, the exit velocity of the jet was increased. In addition, the added superheated steam was found to inhibit the combustion of methane, resulting in a high temperature area and a decrease in the length of the potential core of the jet upon increasing the content of superheated water vapor. Compared with an ambient temperature of 300 K, a higher ambient temperature of 1700 K resulted in a reduction in the density of the free space gas and an enhancement in the diffusion of the central jet, which led to a larger high temperature zone that deviated from the central axis. The jet velocity and the length of the potential core were also extended, and its rate of decay was reduced. Furthermore, it was found that the vorticity curve of the coherent supersonic jet exhibited two peaks. More specifically, the peak close to the axis was influenced by the central jet, and as the distance to the exit increased, the velocity of the central jet decreased; the difference between the velocity and the environment also decreased. As a result, the vorticity decreased, along with the corresponding peak intensity. The second peak was influenced by the combustion reaction of methane in the annular slot. As the distance from the outlet increased, the combustion reaction became more complete, and the combustion area became larger due to diffusion. As such, the corresponding peak became wider, the superheated steam present in the jet inhibited the combustion reaction, and the jet mixed with the air in the reaction environment at an earlier point. Moreover, our results indicated that the distribution of $H_2O$ in the jet was controlled by the coupling of the diffusion and combustion reactions. In a high-temperature environment, the free space density is low, and the jet spreads out to a greater extent. As a result, the high-temperature environment is conducive to the combustion reaction. Overall, the aim of our study was to reduce the generation of smoke

and dust in an electric furnace by mixing the oxygen injection feed and incorporating a high-temperature steam. The current research mainly focused on the characteristics of the mixed jet, including the speed, the temperature, and the turbulent kinetic energy. During our follow-up work, we will further study the effect of mixing oxygen and superheated steam on the stirring effect of the molten pool, the quality of the molten steel, the effect of reducing smoke and dust production, and optimization of the corresponding parameters.

**Author Contributions:** Conceptualization, X.L. (Xin Li) and G.W.; methodology, X.L. (Xin Li); software, X.L. (Xin Li); validation, B.T., R.Z. (Ruimin Zhao) and X.L. (Xinyi Lan); formal analysis, X.L. (Xin Li); investigation, X.L. (Xin Li); resources, X.L. (Xin Li); data curation, X.L. (Xin Li); writing—original draft preparation, X.L. (Xin Li); writing—review and editing, X.L. (Xin Li); visualization, X.L. (Xin Li); supervision, R.Z. (Rong Zhu); project administration, X.L. (Xin Li); funding acquisition, R.Z. (Rong Zhu). All authors have read and agreed to the published version of the manuscript.

**Funding:** The authors would like to express their thanks for the support by the National Nature Science Foundation of China (No. 52004023 and No. 52074024) and the Major Science and Technology Innovation Project of Shandong Province (No. 2019JZZY010358).

**Data Availability Statement:** Not applicated.

**Conflicts of Interest:** The authors declare no conflict of interest.

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
