# Peer review of "Study on the Characteristics of Coherent Supersonic Jet with Superheated Steam"

_metals, doi:10.3390/met12050835_

Round 1

Reviewer 1 Report

This is an interesting paper whose subject is appropriate for publication, but my primary concern is whether the authors’ interpretation of the physics is appropriate.  They consider a highly complex 3D reacting jet with different mixtures of gases and different background fluids but do not check the simple things such as, how does changing the fluid in an isolated, axisymmetric Laval nozzle change its expansion structure.  Although the authors speculate on how adding H2O and changing the temperature of the ambient fluid changes combustion, some simple axisymmetric calculations without combustion could provide a much clearer understanding of the physics.  In addition, they compute a full 360 degree view of a problem that appears to be circumferentially periodic.  In the following, I give more detail in these issues followed by some minor editorial points.

The length of the potential core appears to be the most significant quantity of interest in the paper but there are so many variables present it is difficult to determine which is controlling.  Some specific issues:

  1. The flow through a Laval nozzle is dependent on gamma. As more and more H2O is mixed with the O2, gamma changes causing the nozzle to flow off-design resulting in an under- or over-expanded jet.  This effect should be documented by computing the nozzle flowfield as a stand-alone geometry to compare the 100% O2 case with the 25% O2/75% H2O.  The degree of over/under-expansion will have a significant effect on the potential core.  Is the primary effect because H2O causes the nozzle to operate off-design, or because H2O affects the combustion?  
  2. How would the core length change with H2O addition in a simple axisymmetric geometry without the shrouding combustion?
  3. What role does the shear layer play in determining the potential core, and how is the shear layer (or its equivalent) affected by combustion?
  4. Can you verify whether changing the temperature (density?) of the ambient gas changes the combustion rate? In the absence of the combustion shroud, I would expect the pressure of the ambient gas to be the primary effect, although a lower density fluid would change the shear layer mixing (item 3 above).
  5. How does combustion affect the potential core length (compare a simple axisymmetric jet with the present computations).

Show axisymmetric computation (without external shroud jets or combustion) for 0% and 74% H2O to demonstrate the effect of gamma on the flowfield.  The ‘potential core length’ is a strong function of the nozzle design/off-design operation.  Ignoring friction, the potential core length of a nozzle at design back pressure is infinite.  In a real fluid the potential core length is determined by how rapidly the surrounding shear layers mix.  When the nozzle is over- or under-expanded, the potential core length (as defined as 95% decay) will be finite even for an inviscid jet.  It is the degree of under/over expansion that determines it.

The experiment clearly shows the geometry is periodic in the circumferential direction.  Why was not a 1/10th sector used instead of the entire domain.  Steady RANS simulations should not produce any asymmetries between sectors and reducing the domain to 1/10th the size would give much needed resolution.

I understand the geometry is complex, but a simple line drawing is needed to give an overall view of the problem.  I suggest combining the nozzle detail in the left of Fig. 4 with a simple line drawing as a first description of the computational domain.  The details of the three rings of holes could then be discussed. 

My difficulties with the present Fig 3 are that there is no indication of flow direction (upper right to lower left?), but region outside the nozzle is labeled as ‘outlet’.    From the experiment I would expect this would be a wall but if it is not a wall it should be an inlet (and the BCs should be specified).  Also, I do not understand the significance of the light blue region (this is a wall and the remainder is an outflow (inflow?) boundary?).  Why there is a rectangular region around the upper right picture.  A simple line drawing would make these details much clearer.  What is the ‘epoxy inlet (and epoxy is also mentioned in lines 45, 314, 317).  Also, the ring of holes in the experiment appear to be cylindrical, but the computation shows a much different shape.  I presume this is done to simplify grid generation, but it should be so stated.

It is not stated, but I assume all computations were done with the shrouding jets and reaction (but a comparison with and without combustion and surrounding jets is crucial to understanding the physics).  

Local regions around the nozzle exit need to be shown.  A cross-sectional view of Mach number or other variable would provide better understanding of the observed fluctuations and give details of the off-design operation, shear layer growth, etc.  The overall views fail to show which physics are controlling.

Some minor comments:

The density in a Laval nozzle with the stated area ratios changes by a factor of four from the inlet to the exit, yet in Fig. 7 the density through the nozzle is essentially constant?

The pitot tube equations do not specify how the value of gamma was determined for the experiments—was it changed when going from O2 to the O2/H2O mixtures?

Eq. 11 includes TKE generation due to buoyancy.  Buoyancy should not have been a factor in the present calculations.  If so, omit this term in the equation.

The diameter of the computational domain is incorrectly labeled in Fig. 4 as 70 mm.

The caption in Fig 5 should state ‘axial velocity distribution on the centerline’.  

It should be stated that x = 0 is the nozzle exit plane.

Adding combustion around the jet may affect the (perceived) back pressure and bring the jet closer or farther from design conditions.

Need to distinguish between the effects of back pressure and gamma and shear layers.

Show a close-up view of the region near the nozzle to see details; 

Why is the density in the outer O2 ring so much lower than the nozzle? 

The furnace diameter is 1200 mm and its length is 4000mm.

What value of gamma is used in Eqs. 1 and 2? And how determined?

Paper says a modified k-e model was used, but gives equations for ‘standard’ k-e.

Throat diameter:  27; exit D 35, design Mexit 2.0 inlet D 52

Area ratios:  the density should change by a factor of 4 through the nozzle, but the results show it is nearly constant?

Epoxy: line 45, 314, 317

Distance misspelled in upper left Fig. 10

CO2 and H2O distributions are (nearly) correlated only one plot should be sufficient?

Reviewer 2 Report

The computational fluid characteristics of coherent supersonic jet with superheated steam were well examined in this report.

Please check the following matters.

Please write non-abbreviation word for the first appearance in the manuscript.

k-e turbulence or K-e model

In Figure 1 or 2, please indicate the size of coherent jet lance.

In Figure 2, please show the scale is mm.

In equation (1), po →po

In equation (5), vT →vT

In Figure 5. X axis,  Z/m →Z(m)  like Y axis. Or every unit in figures write  /.

Please unify the writing style of unit.

In Figure 6. Kg/m3  →3 is superscript.  O2 →2 is subscript.

Please explain the inlet speed is very high comparing with other figures.

In Figure 10, 13 ,15.  H2O →2is subscript.

Are there no experimental plots?

In Figure 16. CO2 →2is subscript.

Please write the main prospect points to reduce the smoke and dust in an electric furnace in the discussion section.

Round 2

Reviewer 1 Report

The authors have made only cosmetic changes.  I have spent far more time reviewing the manuscript than they spent revising it.  My primary comment is that there is still no discussion of the dominating physics.  Most of the observations in the paper state that the computational results show that changing this parameter makes the potential core length longer or shorter.  There is no discussion as to what physics are inducing the changes. 

There are three key processes that control the potential core length: 

  1. The degree to which the nozzle is operating off-design,

Changing the fluid in the nozzle (adding H2O) changes the gamma of the fluid and causes a nozzle designed for pure O2 to operate off design.  The close-up plots in Fig. 7 show that the velocity inside the supersonic portion of the nozzle is considerably faster with 75% H2O than with 0% H2O,  Further, the external jet is nearly ideally expanded with 0% H2O whereas with 75% H2O it is substantially under-expanded.  This clearly shows that H2O causes the nozzle to operate in an off-design fashion.  Operating a nozzle in off-design conditions has a major effect on the potential core length and this effect should be acknowledged/discussed in the paper.  A more appropriate analysis would re-design the nozzle to operate with the current fluid (i.e., the % H2O) so that the effect of off-design operation could be distinguished from the effect H2O on combustion.

Figure 6 shows that without combustion adding 75% H2O to the jet increases the potential core length by about 20%.  This is totally due to off-design operation.  Using a similar off-design nozzle with pure O2 would also increase the potential core length.  Interestingly, the presence of combustion appears to decrease the effect of off-design to about 10 %.

  1. The rate of growth of the turbulent shear layer at the periphery of the jet

The shear layer growth rate is dependent on the density (temperature) of the ambient fluid.  Lower density (higher temperature) decreases the growth rate and conversely. 

  1. The combustion in the fluid in the shroud.

Adding water to the nozzle flow will have a detrimental effect on the rate of combustion, though this effect could be largely mitigated by increasing the O2 n the inner and outer rings.

The off-design operation induced by adding H2O is never mentioned.  Shear layer effects are alluded to, but plots showing their relative effect on the core length are absent.  The effect of H2O on combustion is mentioned, but methods for counteracting this by changing the flow rates in the surrounding O2/CH4 jets is not mentioned.  phenomena are not acknowledged apart from saying ‘there was less combustion’ or ‘increasing the external temperature enhances the reaction’.

The reason for computing the entire domain rather than a 1/10th sector are not discussed.  As I noted before, running a sector would decrease CPU costs by a factor of 10, or would allow the grid to be doubled in all three directions without impacting CPU time.

The density inside the nozzle should decrease by a factor of 4 from inlet to exit, yet in Fig. 8 it appears to be constant.  The velocity profiles in Fig. 6 and 7 show no evidence of a recirculation region behind the base region.  Are the results in Fig. 8 with or without combustion?

The velocity fluctuations on the center line just downstream of the nozzle exit plane arise because the nozzle is not perfectly designed and their increasing magnitude with addition of H2O is clear.  Also, the fluctuations are identical with both back pressures—proof that the fluctuations are inviscid in nature and are unaffected by the ambient fluid.

Figure 9 would be much more understandable if there were a corresponding picture showing how rapidly the shear layer reaches the centerline.

It should be stated in the longitudinal section plots whether these plots are taken at the centerline of the O2/CH4 ports or between them.  Similarly for the radial plots.

Some minor comments:

Figure 1   to what dimension does 1200 mm refer?

Eq. 1 and 2:  How is gamma determined, what is the recovery factor?

Line 145  the words ‘is the fluid velocity with’ should not be deleted?

Line 146  the ‘T’ should be a superscript (transpose)

The buoyancy term from Eq. 11 but it still appears in 14

Line 270 has some additional words that do not belong
